# The Novel ALG-2 Target Protein CDIP1 Promotes Cell Death by Interacting with ESCRT-I and VAPA/B

**DOI:** 10.3390/ijms22031175

**Published:** 2021-01-25

**Authors:** Ryuta Inukai, Kanako Mori, Keiko Kuwata, Chihiro Suzuki, Masatoshi Maki, Terunao Takahara, Hideki Shibata

**Affiliations:** 1Department of Applied Biosciences, Graduate School of Bioagricultural Sciences, Nagoya University, Furo-cho, Chikusa-ku, Nagoya 464-8601, Japan; inukai.ryuta@e.mbox.nagoya-u.ac.jp (R.I.); kanako.mkhn8@gmail.com (K.M.); chikama.three@gmail.com (C.S.); mmaki@agr.nagoya-u.ac.jp (M.M.); takahara@agr.nagoya-u.ac.jp (T.T.); 2Institute of Transformative Bio-Molecules (WPI-ITbM), Nagoya University, Furo-cho, Chikusa-ku, Nagoya 464-8601, Japan; kuwata@itbm.nagoya-u.ac.jp

**Keywords:** adaptor, calcium-binding protein, cell death, ESCRT-I, *PDCD6*, protein-protein interaction, VAPA, VAPB

## Abstract

Apoptosis-linked gene 2 (ALG-2, also known as PDCD6) is a member of the penta-EF-hand (PEF) family of Ca^2+^-binding proteins. The murine gene encoding ALG-2 was originally reported to be an essential gene for apoptosis. However, the role of ALG-2 in cell death pathways has remained elusive. In the present study, we found that cell death-inducing p53 target protein 1 (CDIP1), a pro-apoptotic protein, interacts with ALG-2 in a Ca^2+^-dependent manner. Co-immunoprecipitation analysis of GFP-fused CDIP1 (GFP-CDIP1) revealed that GFP-CDIP1 associates with tumor susceptibility gene 101 (TSG101), a known target of ALG-2 and a subunit of endosomal sorting complex required for transport-I (ESCRT-I). ESCRT-I is a heterotetrameric complex composed of TSG101, VPS28, VPS37 and MVB12/UBAP1. Of diverse ESCRT-I species originating from four VPS37 isoforms (A, B, C, and D), CDIP1 preferentially associates with ESCRT-I containing VPS37B or VPS37C in part through the adaptor function of ALG-2. Overexpression of GFP-CDIP1 in HEK293 cells caused caspase-3/7-mediated cell death. In addition, the cell death was enhanced by co-expression of ALG-2 and ESCRT-I, indicating that ALG-2 likely promotes CDIP1-induced cell death by promoting the association between CDIP1 and ESCRT-I. We also found that CDIP1 binds to vesicle-associated membrane protein-associated protein (VAP)A and VAPB through the two phenylalanines in an acidic tract (FFAT)-like motif in the C-terminal region of CDIP1, mutations of which resulted in reduction of CDIP1-induced cell death. Therefore, our findings suggest that different expression levels of ALG-2, ESCRT-I subunits, VAPA and VAPB may have an impact on sensitivity of anticancer drugs associated with CDIP1 expression.

## 1. Introduction

Apoptosis-linked gene 2 (ALG-2, gene symbol *PDCD6*) is a 22-kDa protein having five repetitive EF-hand motifs called the penta-EF-hand (PEF) domain [1]. It forms a homodimer or a heterodimer with its paralogous protein Peflin and interacts Ca^2+^-dependently with a variety of target proteins that function in multifaceted cellular processes including apoptosis, cancer development, signal transduction, membrane trafficking, and post-transcriptional control (see [2] for a review). A large body of structural and biochemical evidence suggests Ca^2+^-sensitive adaptor functions of ALG-2, particularly in the early secretory pathway [3]. ALG-2 is recruited to the endoplasmic reticulum exit site (ERES) by direct interaction with Sec31A, a component of the outer layer of coat protein complex II (COPII) [4,5,6], and bridges between Sec31A and annexin A11 to stabilize Sec31A at the ERES [7]. Whereas the ALG-2-Peflin heterodimer functions as an adaptor for the ubiquitin ligase CUL3-KLHL12 to ubiquitinate Sec31A [8], the ALG-2 homodimer interacts with and polymerizes Trk-fused gene (TFG) protein to concentrate TFG at the ERES [9]. ALG-2 also links MAPK1-interacting and spindle-stabilizing-like (MISSL) and microtubule-associated protein 1B (MAP1B), which is likely to be required for efficient transport of conventional cargo proteins [10]. In the endocytic pathway, ALG-2 bridges two target proteins, ALG-2-interacting protein X (ALIX, also known as AIP1 and HP95) and endosomal sorting complex required for transport (ESCRT)-I [11,12]. Among the growing list of functions of ESCRT in cellular processes related to membrane remodeling, ALG-2 functions as an activator of ALIX in the multivesicular body sorting pathway [13] and as an ESCRT-0 in plasma membrane repair [14,15,16].

Although the murine gene encoding ALG-2 was first reported to be essential for execution of apoptosis triggered by T-cell receptor engagement in mouse T-cell hybridoma 3DO [17], T-cells from ALG-2-deficient mice generated by gene targeting are as susceptible as wild-type T-cells to apoptotic stimuli [18]. Nevertheless, potential roles of ALG-2 and its target proteins in multiple cell death pathways have been reported. In a series of studies, Sadoul and colleagues revealed that the ALG-2 binding region of the C-terminal Pro-rich region in ALIX is crucial for ALIX-mediated neuronal cell death [19,20,21]. They also demonstrated that ALIX is involved in tumor necrosis factor-α (TNF-α)-induced cell death and that an ALIX mutant with deletion of the ALG-2 binding region has a protective effect against cell death [22]. In addition, the ALG-2 binding region of ALIX is required for activation of caspase-9 by ALIX in thapsigargin-induced apoptosis [23]. ALIX has been reported to be upregulated in the neuronal soma during kinetic acid-induced neuronal death in the hippocampus and during neurodegeneration in the striatum in model rats of Huntington’s disease [24,25]. In addition to ALIX, some proteins that interact with ALG-2 are also upregulated in response to DNA damage. Those proteins include the p53-inducible proapoptotic protein Scotin and the death-associated protein kinase DAPK1 [26,27]. Whereas overexpression of ALG-2 stabilizes overexpressed Scotin [26], co-expression of ALG-2 and DAPK1 synergize on execution of apoptosis [27]. However, the molecular mechanisms underlying promotion of cell death by ALG-2 and the involvement of its adaptor function in cell death pathways remain largely unclear.

Based on results of structural, mutational and biochemical analyses of ALG-2 and its target proteins, ALIX [28], PLSCR3 [29], Sec31A [30] and IST1 [31], we have proposed at least three types of Pro-based ALG-2 binding motifs (ABMs) [2]. In silico screening based on type 1 and type 2 ABMs (ABM-1 and ABM-2) and biochemical far-Western blotting analysis using recombinant ALG-2 revealed new candidate proteins for direct interaction with ALG-2 in the presence of Ca^2+^ [32]. Of these proteins, lipopolysaccharide-induced tumor necrosis factor-α factor-like (LITAFL; also known as cell death-inducing p53-target protein 1, CDIP1) is a proapoptotic protein with an ABM-2-like motif in its N-terminal Pro-rich region. The human CDIP1 gene was first cloned as *C16ORF5* near a region of chromosome 16 associated with a de novo translocation in a patient with epilepsy and mental retardation [33]. Subsequently, Lee and colleagues characterized this gene product as a proapoptotic protein [34,35,36]. In response to DNA damage, CDIP1 is upregulated in a p53-dependent manner [34]. Overexpressed CDIP1 then induces apoptosis through upregulation of TNF-α and sensitization of cells expressing CDIP1 to TNF-α-induced cell death [34,35]. ER stress also activates expression of CDIP1 in a p53-independent manner [36]. During ER stress, CDIP1 appears to trigger cell death by a different pathway involving B-cell-receptor-associated protein 31 (BAP31). CDIP1 interacts with BAP31 at the ER membrane, which requires cleavage of BAP31 and association of the cleaved BAP31 with BAX to induce mitochondria-mediated apoptosis [36].

In this study, we demonstrated that ALG-2 interacts with CDIP1 in a Ca^2+^-dependent manner and that ALG-2 functions as an adaptor bridging CDIP1 and ESCRT-I. CDIP1-induced cell death was enhanced by ALG-2 and ESCRT-I. Furthermore, we identified vesicle-associated membrane protein-associated protein (VAP) A and VAPB as interacting partners of CDIP1. Mutational analysis revealed that the C-terminal two phenylalanines in an acidic tract (FFAT)-like motif is required not only for interaction with VAPA and VAPB but also for the cell death-inducing activity of CDIP1.

## 2. Results

### 2.1. Ca^2+^-Dependent Interaction of ALG-2 with CDIP1

CDIP1 consists of an N-terminal region rich in Pro and a C-terminal LITAF domain (also known as SIMPLE-like domain) responsible for a membrane anchor [37] (Figure 1A). The N-terminal Pro-rich region has a sequence, ^62^PQPGF, similar to the type 2 ALG-2 binding motif (ABM-2) of PLSCR3 and Sec31A (conserved residues underlined) [29,30]. We have reported that biotin-labeled recombinant ALG-2 binds directly to GFP-fused CDIP1 (GFP-CDIP1) in a far-Western experiment in the presence of CaCl_2_ (100 μM) [32], but Ca^2+^-dependency of the interaction remains to be established. In order to address this issue, GFP-CDIP1 was expressed in HEK293 cells and the proteins in the cleared lysate (Input) were immunoprecipitated with a recombinant nanobody against GFP in the presence of the Ca^2+^ chelator EGTA or CaCl_2_. In this experiment, the concentration of CaCl_2_ was set to the same value of 100 μM as for the far-Western experiment [32]. As shown in the upper panel of Figure 1B, Western blot (WB) analysis with a mouse monoclonal antibody against GFP revealed comparable WB signals in the immunoprecipitation (IP) products of GFP and GFP-CDIP1 in the presence of EGTA and CaCl_2_. Endogenous ALG-2 was detected in the IP product of GFP-CDIP1 in the presence of CaCl_2_ but not in the presence of EGTA (Figure 1B, lower panel). This result indicates that the interaction of ALG-2 with CDIP1 is Ca^2+^-dependent.

When protein samples in the IP product of GFP-CDIP1 were resolved by SDS-PAGE, doublet bands and faint multiple slower migrating bands were detected by the antibody against GFP (Figure 1B, upper panel, arrows and asterisk). To examine the possibility of ubiquitination of GFP-CDIP1, proteins in the IP product were resolved in three lanes of an SDS-PAGE gel and the blotted membrane was cut into halves in the middle of the second lane. The left and right membranes were probed with monoclonal antibodies against ubiquitin (Ub) and GFP, respectively (Figure 1C). Multiple slower migrating bands were detected with the antibody against ubiquitin with a pattern of bands similar to that of bands detected with the antibody against GFP, indicating that they correspond to GFP-CDIP1 conjugated with ubiquitin (Figure 1C, asterisk).

To validate the importance of an ABM-2-like sequence in the interaction of CDIP1 with ALG-2, we investigated the effects of deleting five amino acids from residues 62 to 66 (Mut1) (Figure 1A). As shown in the lower panel of Figure 1D, the signal for binding to ALG-2 was significantly reduced for the Mut1 mutant. The result suggests the importance of the ABM-2-like motif for the Ca^2+^-dependent interaction of CDIP1 with ALG-2.

To investigate whether the interaction of ALG-2 with CDIP1 occurs between endogenously expressed proteins, we first examined the expression level of CDIP1 with a commercially available antibody against CDIP1. Since CDIP1 is an ER-stress response protein [36], we treated HEK293 cells with the ER stress agent brefeldin A (BFA), which disrupts transport between the ER and Golgi compartments. As shown in the middle panel of Figure 2A, expression of CDIP1 was gradually increased after treatment with BFA, whereas the expression level of ALG-2 was not altered (Figure 2A, upper panel). Appearance of the major double bands seems due to post-translational modifications other than ubiquitinations (Figure 1C). We then performed a co-immunoprecipitation assay by immunoprecipitating endogenous ALG-2 from the cell lysate of HEK293 cells treated with BFA for 9 h (Figure 2B). In our assay condition, ALG-2 was immunoprecipitated by our antibody against ALG-2 with comparable efficiency in the presence of EGTA or CaCl_2_ (Figure 2B, upper panel). In this experiment, we lowered the concentration of CaCl_2_ in the binding buffer (10 μM) to prevent aggregation of ALG-2 during the binding assay. WB signals of CDIP1 were detected as doublet bands in the IP product of ALG-2 in the presence of CaCl_2_ but not in the presence of EGTA. These results suggest that the level of CDIP1 is elevated in response to ER stress and that the CDIP1 protein interacts with ALG-2 in a Ca^2+^-dependent manner.

### 2.2. Promotion of CDIP1-Induced Cell Death by ALG-2

During the course of the experiment using HEK293 cells expressing GFP-CDIP1, we noticed that overexpression of GFP-CDIP1 appeared to be toxic to the cells. Since CDIP1 is considered to be a proapoptotic protein [34,36], we investigated the role of ALG-2 in CDIP1-induced cell death by using previously established ALG-2 knockout (KO) HEK293 cells [38]. For quantifying cell death, the extracellular activity of lactate dehydrogenase (LDH) released from the cytosol of damaged cells was measured and was normalized to the amount of total proteins in the cell lysate. To monitor the expression levels of exogenously expressed GFP-CDIP1 and ALG-2, the protein in the cell lysate was resolved by SDS-PAGE, followed by WB with antibodies against GFP and ALG-2 (Figure 3A). As shown in Figure 3B, overexpression of either GFP alone or both GFP and ALG-2 had no significant effect on cell death assessed by monitoring LDH activity (lanes 2 and 3). In contrast, overexpression of GFP-CDIP1 promoted LDH release (lane 4), indicating cytotoxicity of GFP-CDIP1. Cell death was further enhanced by co-expression of ALG-2 (lane 5). This result suggests that ALG-2 is not required for cell death induced by overexpression of CDIP1 but facilitates CDIP1-induced cell death. To further characterize cell death induced by overexpression of CDIP1, we investigated whether cells expressing GFP-CDIP1 undergo necroptic cell death. We analyzed phosphorylation of Mixed lineage kinase domain-like protein at Ser 358 (p-MLKL), a well-accepted marker of necroptosis [39] (Figure 3C). In ALG-2 KO HEK293 cells, WB signal for MLKL was clearly detected (lower panel, lane 1). In contrast, a faint band similar in size to that of MLKL was detected by the antibody against phosphorylated MLKL in ALG-2 KO HEK293 cells (upper panel, lane 1), in agreement with little release of LDH in Figure 3B. The intensity of the band was not significantly changed in cells expressing GFP-CDIP1 (lane 4) or in cells co-expressing GFP-CDIP1 and ALG-2 (lane 5). Next, to assess the participation of caspases in GFP-CDIP1-induced cell death, we measured intracellular caspase-3/7 activity. As shown in Figure 3D, overexpression of GFP-CDIP1 resulted in a significant increase in caspase-3/7 activity. This increase tended to be greater in cells co-expressing GFP-CDIP1 and ALG-2. These results indicate that GFP-CDIP1-induced cell death is attributed to caspase 3/7-mediated cell death.

### 2.3. Ca^2+^-Dependent Adaptor Function of ALG-2 to Bridge CDIP1 with ESCRT-I

On the molecular basis for the involvement of ALG-2 in CDIP1-induced cell death, we considered the possibility that ALG-2 may mediate the physical association of CDIP1 with a specific binding protein by the adaptor function of ALG-2 [2,3]. To test this possibility, the IP product of GFP-CDIP1 in the presence of either EGTA or CaCl_2_ was immunoblotted with antibodies against known ALG-2-binding proteins including Peflin, ALIX, annexin A11 (ANXA11), MISSL, TSG101, Sec31A and TFG (Figure 4A). WB signals for Peflin and TSG101, but not for other proteins, were detected in the IP product of GFP-CDIP1 in the presence of Ca^2+^. In the presence of EGTA, a faint signal for TSG101 was only detectable in the IP product of GFP-CDIP1. Thus, CDIP1 appears to interact with TSG101 even in the absence of Ca^2+^, and ALG-2 likely supports the association of CDIP1 with TSG101 in the presence of Ca^2+^ via its adaptor function. In order to check this possibility, GFP-CDIP1 was expressed in ALG-2 KO HEK293 cells and immunoprecipitated with a nanobody against GFP in the presence of EGTA or CaCl_2_ (Figure 4B). A faint signal for TSG101 was detected with similar intensities in the IP product of GFP-CDIP1 in the presence of EGTA and CaCl_2_ (Figure 4B, lanes 9 and 10). When ALG-2 was co-expressed with GFP-CDIP1 in ALG-2 KO cells, the co-immunoprecipitated TSG101 was increased by the expression of ALG-2 in the presence of CaCl_2_ but not in the presence of EGTA (Figure 4B, lanes 11 and 12). The result indicates that ALG-2 bridges between CDIP1 and TSG101 in a Ca^2+^-dependent manner.

CDIP1 and Lipopolysaccharide-Induced TNF-α Activating Factor/Small Integral Membrane Protein of Lysosome/late Endosome (LITAF/SIMPLE, also known as PIG7) belong to a class of monotopic integral membrane proteins and share a similar domain structure [37]. LITAF/SIMPLE was reported to interact with TSG101 via its PSAP sequence [40]. To compare the binding avidity of CDIP1 and that of LITAF/SIMPLE to TSG101 and the effects of Ca^2+^ on their binding, GFP-LITAF was expressed in HEK293 cells and an IP assay was conducted (Figure 4C). In the presence of EGTA, the signal intensity of the TSG101 band obtained by co-IP with GFP-LITAF/SIMPLE was higher than that obtained with GFP-CDIP1 (Figure 4C, lanes 6 and 8). In contrast, in the presence of CaCl_2_, GFP-CDIP1 co-immunoprecipitated TSG101 more efficiently than did GFP-LITAF/SIMPLE (Figure 4B, lanes 7 and 9). The TSG101 band in the IP product of GFP-LITAF/SIMPLE in the presence of CaCl_2_ had an intensity equivalent to that in the presence of EGTA, indicating no effect of Ca^2+^ on the interaction between LITAF/SIMPLE and TSG101. Being consistent with the lack of ALG-2 binding motifs in LITAF/SIMPLE, there was no ALG-2 band in the IP product of GFP-LITAF/SIMPLE in both conditions (Figure 4C, lanes 8 and 9).

TSG101 functions as a core subunit of the heterotetrameric endosomal sorting complex required for transport-I (ESCRT-I), together with single copies of VPS28, VPS37 and MVB12/UBAP1 [41]. ESCRT-I exists as multiple isocomplexes due to the presence of multiple isoforms of individual subunits. For example, there are four isoforms of human VPS37 (VPS37A, VPS37B, VPS37C and VPS37D) and at least three isoforms of MVB12/UBAP1 (MVB12A, MVB12B and UBAP1) (Figure 5A). The four isoforms of VPS37 share the Mod(r) domain mediating interaction with the remaining subunits of ESCRT-I [42,43,44]. VPS37A has a UEV domain in its N-terminus, whereas VPS37B, VPS37C, and VPS37D have Pro-rich regions with different types of protein-protein interaction motifs in their C-termini. Previously, we demonstrated direct interaction of ALG-2 with VPS37B and VPS37C but not with VPS37A or VPS37D [12]. To test whether CDIP1 associates with ESCRT-I machineries with different preferences for specific combinations of the isoforms, we performed an IP assay by expressing GFP-CDIP1 and each of four sets of 12 different combinations of Myc-tagged ESCRT-I subunits (TSG101, VPS28, each of four VPS37 isoforms, and each of three MVB12/UBAP1 proteins) in HEK293 cells. In this experiment, we chose a physiologically reasonable concentration of Ca^2+^ (1 μM). As shown in Figure 5B–D, in the presence of CaCl_2_, WB signals for ALG-2 were detected in the IP product of GFP-CDIP1 from cells co-expressing all combinations of four VPS37 isoforms and three MVB12/UBAP1 proteins. No WB signal for Myc-MVB12A was detected in the IP products from cells co-expressing any combination of VPS37 and MVB12/UBAP1 in the presence of both EGTA and CaCl_2_ (Figure 5B). WB signals for other subunits were detected in the immunoprecipitates, probably because they could form ESCRT-I isocomplexes containing endogenous MVB12B or UBAP1, but not Myc-MVB12A, in cells. In contrast, all Myc-tagged subunits in ESCRT-I isocomplex 37B/12B (expressing Myc-VPS37B and Myc-MVB12B) and ESCRT-I isocomplex 37C/12B (expressing Myc-VPS37C and Myc-MVB12B) were co-immunoprecipitated with GFP-CDIP1 in the presence of CaCl_2_ (Figure 5C). Although a WB signal for Myc-VPS28 was barely detected, faint signals for other Myc-tagged subunits in ESCRT-I isocomplex 37D/12B (expressing Myc-VPS37D and Myc-MVB12B) were seen in the IP in the presence of EGTA and CaCl_2_. Expression levels of the ESCRT-I isocomplex including Myc-UBAP1 were relatively low under our experimental conditions, but the isocomplex 37B/UBAP1 (expressing Myc-VPS37B and Myc-UBAP1) was co-immunoprecipitated with GFP-CDIP1 in the presence of CaCl_2_ (Figure 5D). WB signals of isocomplex 37D/UBAP1 (expressing Myc-VPS37D and Myc-UBAP1) were seen in the IP products in both conditions. Collectively, the results suggest that CDIP1 associates preferentially with ESCRT-I isocomplexes 37B/12B, 37C/12B and 37B/UBAP1 in a Ca^2+^-dependent manner and raise the possibility that CDIP1 forms a complex with ESCRT-I isocomplex 37D/UBAP1 regardless of the presence of Ca^2+^.

### 2.4. Promotion of CDIP1-Induced Cell Death by ESCRT-I

To examine the role of ESCRT-I in cells expressing CDIP1, an LDH assay was performed. Either GFP or GFP-CDIP1 was co-expressed with each of the ESCRT-I isocomplexes in HEK293 cells (Figure 6A) and the released LDH activity from the cells was measured (Figure 6B). ESCRT-I isocomplexes containing MVB12B were examined to assess the effect of differential binding of CDIP1 to ESCRT-I isocomplexes containing each of the VPS37 isoforms on cell death. Overexpression of GFP with or without any ESCRT-I isocomplexes had no effect on the release of LDH into the culture medium (Figure 6B, lanes 2 to 6). Overexpression of GFP-CDIP1 in HEK293 cells caused an increase in LDH leakage, and co-expression of the ESCRT-I isocomplexes with GFP-CDIP1 led to further release of LDH. Co-expression of ESCRT-I isocomplex 37C/12B resulted in a significant increase in LDH leakage compared with co-expression of other ESCRT-I isocomplexes (Figure 6B, lanes 7–11). These results indicate that overexpression of ESCRT-I has no cytotoxicity in cells expressing little CDIP1 proteins and suggests that CDIP1-induced cell death can be promoted to different degrees by ESCRT-I isocomplexes.

### 2.5. Identification of VAPA and VAPB as Target Proteins for CDIP1

To gain more insights into the molecular mechanisms underlying CDIP1-induced cell death, we tried to identify uncharacterized interacting proteins for CDIP1. For this purpose, we established an MCF-7-derived cell line expressing tetracycline-inducible CDIP1. A WB signal for CDIP1 was detected as doublet bands in the cleared cell lysate from cells treated with the tetracycline analogue doxycycline (DOX) but not from the untreated cells (Figure 7A, upper panel, lanes 1 and 2). To test whether the CDIP1 inducible system could be used for identification of novel interacting proteins, we first examined IP assay, in which IP products were prepared from cells treated or not treated with DOX by using an antibody against CDIP1 or a control antibody. Bands for CDIP1 were seen in the IP product obtained by the CDIP1 antibody from cells treated with DOX but not from cells not treated with DOX. The IP product obtained by using a control antibody from cells treated with or not treated with DOX did not contain CDIP1. A WB signal for ALG-2 was detected in the IP product obtained by the antibody against CDIP1 from cells treated with DOX but not in the IP product from non-treated cells or from treated cells by using a control antibody (Figure 7A, middle panel). In the lower panel of Figure 7A, WB signals for heavy chains of immunoglobulin G (IgG–H) were detected in the four IP products with similar intensities (indicated by asterisk). In contrast, a WB signal for TSG101 was only seen in the IP product obtained by the antibody against CDIP1 from cells treated with DOX as the slightly faster migrating band (indicated by arrow) than the bands for IgG-H. The results reflect the interaction of CDIP1 with ALG-2 and TSG101 and thus indicate that the inducible cell line and the antibody against CDIP1 are useful tools for searching for proteins interacting with CDIP1. To identify proteins with various sizes that interact with CDIP1, as shown in Figure 7B, we screened candidate proteins from immunoprecipitates with the anti-CDIP1 antibody followed by mass spectrometry. By searching the Mascot database on the basis of the LC-MS/MS data, CDIP1, ALG-2, Peflin, BAP31, VAPA and VAPB were identified with high confidence (PSMs > 30, more than four unique peptides) (Figure 7C). BAP31 was previously characterized as a protein that interacts with CDIP1 [36]. To confirm whether VAPA and VAPB are associated with CDIP1, we independently prepared immunoprecipitates obtained by the anti-CDIP1 antibody from cleared lysate of the DOX-treated cells and analyzed them. As shown in Figure 7D, WB signals for VAPA were detected in the IP product of the antibody against CDIP1 but not in that of the control antibody. The intensities of the VAPA signal in the IP product of the antibody against CDIP1 were unchanged under conditions of different Ca^2+^ concentrations in immunoprecipitation without the addition of EGTA or CaCl_2_ and with the addition of EGTA or CaCl_2_. WB signals for VAPB were also stronger in the IP product of the antibody against CDIP1 than in that of the control antibody. These results indicate Ca^2+^-independent interaction of CDIP1 with VAPA and VAPB.

A large number of cytosolic proteins have so far been identified as interacting proteins for VAP proteins, and many of them contain two phenylalanines (FF) in an acidic tract (FFAT)-related motif. The original definition of this motif is EFFDAXE (X: any amino acids) [46]. Subsequent analyses have revealed that substitutions of two or more positions of the defined six amino acids are tolerated, and the number of possible FFAT-related motifs is therefore large (see [47] for review). We searched for candidate motifs in the CDIP1 sequence. There was no typical FFAT motif in CDIP1, but we identified a short sequence in the C-terminal cytoplasmic tail that partially resembled that in the FFAT-related motif in Protrudin [48]. The main difference between FFAT-related sequences in CDIP1 and Protrudin and those in canonical FFAT motifs is that the third phenylalanine residue is replaced by a lysine residue (Figure 8A). We examined whether deletion of the FFAT-like region in CDIP1 has any effect on binding to VAPs. HEK293 cells were transfected with three expression plasmids for GFP-CDIP1 or its mutant, Myc-tagged VAPA and VAPB. Then the cleared cell lysates were used for an IP assay with a nanobody against GFP. As shown in Figure 8B, WB signals for Myc-VAPA and -VAPB were detected in the IP product of GFP-CDIP1 but not in that of GFP. The removal of six amino acids from residues 186 to 191 (Mut2) resulted in complete loss of binding to Myc-VAPA and -VAPB. We further created two individual point mutants and a double mutant (Mut3, F187A; Mut4, D189A; and Mut5, F187A/D189A). While the F187A mutant retained the ability to interact with Myc-VAPA and -VAPB, both the D189A mutant and the double mutant showed no binding to Myc-VAPA and -VAPB. From these results, we concluded that CDIP1 interacts with VAPA and VAPB via an FFAT-like motif, with the fourth aspartate residue in this motif being important for the interactions. In contrast, binding to ALG-2 and TSG101 remained unaffected by the mutations in the FFAT-like region (Figure 8B). We also found that GFP-LITAF/SIMPLE interacted with Myc-VAPA and Myc-VAPB at levels comparable to those of Mut3 of GFP-CDIP1.

### 2.6. Essential Role of the FFAT-Like Motif for CDIP1-Induced Cell Death

To assess the importance of the FFAT-like motif in the cellular function of CDIP1, the cytotoxic effect of the mutant proteins was evaluated by release of LDH from HEK293 cells transiently expressing GFP-CDIP1 and its mutants (Figure 9). Expression of GFP-CDIP1 alone resulted in markedly increased release of LDH. Mutants lacking the ability to interact with VAPs, including Mut2, Mut4 and Mut5, showed no appreciable release of LDH comparable to that of control GFP, whereas Mut3 displaying a reduced avidity for VAPs was still able to induce the release of LDH. Thus, we concluded that the FFAT-like motif in CDIP1 is important for its cell death-inducing activity.

### 2.7. Ternary Complex Formation between CDIP1, VAPB and ALG-2

To examine whether interaction between CDIP1 and VAP proteins occurs at their endogenous expression levels, we performed a co-IP assay of HEK293 cells treated with BFA. As shown in Figure 10A, equivalent WB signals for VAPB were detected in the IP product of an antibody against VAPB, regardless of the presence of EGTA or CaCl_2_. The IP products in both conditions contained CDIP1. This indicates that CDIP1 interacts with VAPB at their endogenous expression levels in a Ca^2+^-independent manner. In contrast, the WB signal for ALG-2 was detected the IP product of VAPB in the presence of CaCl_2_ but not in the presence of EGTA. This suggests that ALG-2 can form a complex with CDIP1 and VAPB in response to Ca^2+^.

## 3. Discussion

ALG-2 was originally identified as a gene product required for apoptosis in mouse T-cell hybridomas [17]. Recently, He et al. reported that ALG-2 interacts with Rpn3/PSMD3, a component of the 26S proteasome, resulting in promotion of the degradation of the BCL-2 family protein MCL1 in apoptotic T-cells [49]. That study highlights the potential pro-apoptotic function of ALG-2 in T-cells. However, the importance of ALG-2 in T-cell receptor-mediated apoptosis is still a controversial issue because no abnormality was observed in T-cells of *ALG-2* KO mice [18]. On the other hand, ALG-2 has been thought to be a possible modulator at the interface between cell proliferation and cell death [50]. Malignant liver tissues and lung tumors exhibit high ALG-2 expression compared to their normal tissues [51,52]. Whereas ALG-2 expression levels were reported to be unrelated to patient survival in breast, colon, and lung cancers [52], ALG-2 has been shown to be a potential prognostic biomarker for predicting clinical efficacy in certain types of cancers. Yamada et al. performed a microarray analysis using forty endoscopic biopsy samples from gastric cancer patients prior to chemotherapy including 5-fluorouracil (5-FU) and 5-FU-based drugs, taxanes, CPT-11 and cisplatin [53]. They found that lower expression levels of ALG-2 mRNA were correlated with poor overall survival. Subsequently, overexpression of ALG-2 in ovarian and advanced gastric cancer cells was reported to show synergic inhibition of cell viability in combination with treatment of anticancer drugs, cisplatin and 5-FU [54,55]. In addition, Briffa et al. characterized a panel of fifteen colorectal cell lines and revealed that ALG-2 expression level was significantly higher in cells that are sensitive to 5-FU, oxaliplatin, and the PI3K/mTOR inhibitor BEZ235 [56]. The results of these studies suggest that ALG-2 may have an important role in killing of cancer cells in response to cytotoxic agents. However, verification is needed by identification of a target protein(s) of ALG-2 in the pathway of cell death. In this study, we identified CDIP1 as a novel target protein for ALG-2 with ABM-2-like motif (Figure 10B). CDIP1 is a pro-apoptotic protein for which expression is upregulated in cells treated with cytotoxic anticancer drugs including etoposide, camptothecin, and BFA [34,36] (Figure 2A). Overexpression of CDIP1 in HEK293 cells induced cell death (Figure 3, Figure 6 and Figure 9), being consistent with previous observations in several other cell lines [34]. ALG-2 was dispensable for CDIP1-induced cell death, but its expression conferred enhanced cell death in cells co-expressing CDIP1 (Figure 3). Therefore, CDIP1 would be a potential target for ALG-2 in cancer patients undergoing chemotherapy. The importance of Ca^2+^ signaling in execution of apoptosis in cancer cells treated with anti-cancer drugs including BFA and cisplatin have been reported [57,58,59,60]. Thus, ALG-2 could function as a proapoptotic protein by interacting with CDIP1 in response to intracellular Ca^2+^ mobilization in cancer cells expressing CDIP1. CDIP1 interacted with the ESCRT-I component TSG101 and this interaction was increased by the adaptor function of ALG-2 (Figure 4). Protective functions of TSG101 against cell death have been reported: TSG101 and ALIX cooperate to recruit ESCRT-III proteins in repairing lysosomal injury and thereby in preventing cell death [60], while TSG101 rescues predisposition to apoptosis in cells depleted of the RING domain–containing E3 ligase Mahogunin Ring finger 1 via interacting with ALIX [61]. In contrast to those observations, expression of ESCRT-I machineries containing TSG101 promoted CDIP1-induced cell death (Figure 6), in which ALIX seemed not to be involved since CDIP1 showed no interaction with ALIX (Figure 4A). Among the various combinations tested, VPS37C and MVB12B-containing ESCRT-I isocomplex was found to be the best combination for preferential interaction with CDIP1 (Figure 5) and the most effective inducer of CDIP1-induced cell death (Figure 6). These ESCRT-I subunits may also be predictive biomarkers of chemotherapy efficacy in cancer patients by combination with ALG-2.

ALG-2 can bind proteins that contain three types of ABM with different affinities [2]. On the other hand, non-canonical ALG-2-binding sequences were reported in Mucolipin-1/TRPML1 [62], HEBP2/SOUL [63,64] and MAP1B [38]. The ABM-2-like motif in the N-terminal Pro-rich region of CDIP1 was found to be important for Ca^2+^-dependent interaction with ALG-2 (Figure 1D). The ABM-2 was first defined as PXPGF (X, any amino acids) based on results of mutational analysis of the PLSCR3 sequence [29]. Thereafter, we newly defined this motif as [PΦ]PX[PΦ]G[FW]Ω (Φ, hydrophobic; Ω, large side chain) on the basis of results of further studies on the crystal structure of the complex between ALG-2 and the ALG-2-binding peptide of Sec31A and results of subsequent mutational analysis [30]. The ABM-2-like motif of human CDIP1 is optimal (Figure 10B), but CDIP1 showed only a weak ability to bind ALG-2 in our previous far-Western analysis using biotin-labeled recombinant ALG-2 as a probe [32]. However, IP assays with exogenously expressing GFP-CDIP1 or endogenous CDIP1 confirmed a conspicuous interaction between ALG-2 with CDIP1 in the presence of Ca^2+^ (Figure 1 and Figure 2). Because CDIP1 bound TSG101 and various combinations of ESCRT-I machinery even in the absence of Ca^2+^ (Figure 4 and Figure 5), ESCRT-I containing multiple ALG-2 binding motifs may compensate the weak binding ability of CDIP1 in response to Ca^2+^. Indeed, the mutations in the ABM-2-like motif markedly reduced but did not completely abolish the interaction with ALG-2 (Figure 1D). In addition, Peflin may be involved in the interaction between ALG-2 and CDIP1. The core five amino acids of the ABM-2-like motif of CDIP1 (PQPGF) are conserved in avians and in mammals (Figure 10B). In zebrafish, the first Pro residue in the core is replaced by a Gln residue, but this protein contains ABM-3 (MP repeat: ^78^-MPMPMP) at the C-terminus of the ABM-2-like motif, implying that the interaction of ALG-2 with CDIP1 may be conserved in vertebrates. We found that immunoprecipitated CDIP1 was ubiquitinated (Figure 1C). The UEV domain of TSG101 has been shown to bind not only the P[S/T]AP motif but also ubiquitin [65]. Therefore, it may directly interact with ubiquitins attached to CDIP1. Recently, we reported that ALG-2 interacts with a negative regulator of store-operated Ca^2+^ entry, SARAF, and blocks its ubiquitination [66]. In the case of CDIP1, the interaction with ALG-2 appeared not to affect the ubiquitination of CDIP1 since a series of slower migrating bands in the immunoprecipitate of CDIP1 was similar in the absence and presence of Ca^2+^ (Figure 1B).

We identified VAPA and VAPB as interacting proteins for CDIP1. VAPs are tail-anchored membrane proteins that reside on the ER membrane (see [47,67] for review). The cytoplasmic major sperm protein (MSP) domains of VAPs recognize FFAT and FFAT-related motifs and have been shown to interact with many proteins that are anchored in or are attached to other membrane-enclosed compartments. This makes VAPs key components of molecular bridges at membrane contact sites between the ER and other organelles. CDIP1 has a hydrophobic region (HR) at the C terminus (Figure 10B). Qin et al. previously demonstrated that the region is monotopically inserted into the membrane and that CDIP1 is localized to CD63- and LAMP1-positive endocytic compartments [37]. Thus, CDIP1 may form contact sites between the ER and the endo/lysosomal organelles via interacting with VAPs. Genome-wide microarray analyses of 50 breast cancer cell lines and 145 primary breast tumors revealed that the VAPB gene is often amplified and, thereby, that its transcript is overexpressed in breast cancer [68,69]. Since the CDIP1 mutants lacking binding avidity to VAPs did not have cell death-inducing activity (Figure 9), it is possible that CDIP1 inhibits the function of VAPB during cell death. Rao et al. reported that VAPB is involved in the promotion of breast tumor growth by elevation of AKT activity that was monitored by its phosphorylation [70]. However, the amount of phosphorylated AKT was not altered by overexpression of CDIP1 in MCF-7 cells under our experimental conditions (data not shown). Although further analysis is needed to address the molecular mechanism underlying the induction of cell death by the complex of CDIP1 and VAPs, ALG-2 and VAPs represent a potential target for interventions that could improve the efficiency of chemotherapy associated with expression of CDIP1.

## 4. Materials and Methods

### 4.1. Antibodies and Reagents

The following commercially available antibodies were used: rabbit polyclonal antibodies against CDIP1 for WB (13824, Cell Signaling Technology, Danvers, MA, USA), CDIP1 for IP (PAB16708, Abnova, Taipei, Taiwan), VAPB (14477-1-AP, Proteintech, Chicago, IL, USA), Peflin (13587-1-AP, Proteintech, Chicago, IL, USA), MISSL (HPA034506, Sigma-Aldrich, St Louis, MO, USA); mouse monoclonal antibodies against GFP (clone B2, sc-9996, Santa Cruz Biotechnology, Santa Cruz, CA, USA), multi-ubiquitin (clone FK2, MBL, Nagoya, Japan), GAPDH (clone 6C5, sc-32233, Santa Cruz Biotechnology, Santa Cruz, CA, USA), Myc (clone 9E10, sc-40, Santa Cruz Biotechnology, Santa Cruz, CA, USA), VAPA (clone 4C12, sc-293278, Santa Cruz Biotechnology, Santa Cruz, CA, USA), TSG101 (clone 4A10, GTX70255, GeneTex, Irvine, CA, USA); goat polyclonal antibodies against ALIX (sc-49268, Santa Cruz Biotechnology, Santa Cruz, CA, USA) and ANXA11 (sc-9322, Santa Cruz Biotechnology, Santa Cruz, CA, USA); rabbit monoclonal antibodies against TFG (ab156866, Abcam, Cambridge, MA, USA), MLKL (14993, Cell Signaling Technology, Danvers, MA, USA), phosphor-MLKL (Ser 358) (91689, Cell Signaling Technology, Danvers, MA, USA). Rabbit polyclonal antibodies against ALG-2 and Sec31A were prepared as described previously [5,7]. Goat antibodies against mouse IgG and rabbit IgG conjugated with horseradish peroxidase (HRP), and rabbit antibody against goat IgG conjugated with HRP were from Jackson ImmunoResearch (West Grove, PA, USA). HRP-conjugated mouse monoclonal antibody against native rabbit IgG (conformation specific) (clone L27A9, 5127, Cell Signaling Technology, Danvers, MA, USA) and HRP-conjugated mouse IgGκ binding protein (sc-516102, Santa Cruz Biotechnology, Santa Cruz, CA, USA) were purchased. Brefeldin A (BFA) and Doxycycline Hydrochloride n-Hydrate (DOX) were purchased from FUJIFILM Wako Chemicals (Osaka, Japan). Puromycin and blasticidin were from InvivoGen (San Diego, CA, USA).

### 4.2. Plasmid Construction

An expression plasmid for SGFP2-fused CDIP1 was constructed as follows: pSGFP2-C3 was obtained by inserting the AgeI and BsrGI-digested SGFP2-encoding cDNA fragment of pSGFP2-C1 (kindly provided by Dr. Wada, Fukushima Medical University School of Medicine, Japan, who constructed the plasmid according to [71] between the AgeI and BsrGI site of pEGFP-C3 (Clontech Laboratories, Palo Alto, CA, USA) by replacement of the EGFP-encoding DNA segment with the DNA fragment encoding SGFP2. A full-length CDIP1-encoding DNA fragment obtained by digesting pEGFP-C3/CDIP1 [32] with EcoRI and SalI was inserted between the EcoRI and SalI sites of pSGFP2-C3. The resultant plasmid was designated pSGFP2-C3/CDIP1. pSGFP2-C3/LITAF was constructed as follows: full-length LITAF (NM_019980.2) was amplified from a HeLa cDNA library (Clontech Laboratories, Palo Alto, CA, USA) by PCR with a pair of primers: forward 5′-cgagctcaagcttcgaattcaaatgtcggttccaggacct-3′ and reverse 5′-gggcccgcggtaccgtcgactacaaacgcttgtaggtgc-3′. The product was inserted between the EcoRI and SalI site of pSGFP2-C3 using a seamless ligation cloning extract (SLiCE) [72]. CDIP1-encoding DNA fragment was obtained by digestion of pEGFP-C3/CDIP1 with HindIII and BamHI, and inserted between the HindIII and BamHI sites of pcDNA3 to construct pcDNA3/CDIP1. pCAG-Myc/UBAP1 was constructed as follows: full-length UBAP1 (NM_001171204.2) was amplified from a HEK293 cDNA library (Clontech Laboratories, Palo Alto, CA, USA) by PCR with a pair of primers: forward 5′-atcagcgaggaggacctgggtaccgcttctaagaagttgggtgctg-3′ and reverse 5′- attaagatatcacccgggggtacctcagctggctcctgcccgag-3′. The product was inserted into the KpnI site of pCAG-Myc/VPS37A [43] using a SLiCE by replacement of the VPS37A-encoding DNA segment with the DNA fragment of UBAP1. pCAG-Myc/VAPA and pCAG-Myc/VAPB were constructed as follows: pOTB7/VAPA and pOTB7/VAPB were obtained from RIKEN Bioresource Center (Tsukuba, Japan) for templates to amplify full-length VAPA and VAPB by PCR, respectively. The primers are: VAPA, forward 5′-atcagcgaggaggacctgggtaccgcgtccgcctcaggggc-3′ and reverse 5′- attaagatatcacccgggggtacctacaagatgaatttccctag-3′; VAPB, forward 5′-atcagcgaggaggacctgggtaccgcgaaggtggagcaggtc-3′ and reverse 5′- attaagatatcacccgggggtacctacaaggcaatcttccc-3′. The products were inserted into the KpnI-digested pCAG-Myc/VPS37A using a SLiCE as described above. pSGFP2-C3/CDIP1 Mut1 (Figure 1A) were obtained by two-step manipulation. First, pEGFP-C3/CDIP1 Mut1 were constructed by PCR-based site-directed mutagenesis using pEGFP/CDIP1 as a template and pair of primers as follows: forward 5′-ccgggtcacccaatgatcccaccacacatg-3′ and reverse 5′-catgtgtggtgggatcattgggtgacccgg-3′. Then CDIP1 mutants-encoding segments were digested by EcoRI and SalI and inserted between the EcoRI and SalI sites of pSGFP2-C3. pSGFP2-C3/CDIP1 Mut2, pSGFP2-C3/CDIP1 Mut3, pSGFP2-C3/CDIP1 Mut4 and pSGFP2-C3/CDIP1 Mut5 were obtained by PCR-based site-directed mutagenesis using pSGFP2-C3/CDIP1 as a template and pairs of primers as follows: primer set for Mut2, forward 5′-cccctgcctcatcactcacacatgccccagc-3′ and reverse 5′-gctggggcatgtgtgattgatgaggcagggg-3′; primer set for Mut3, forward 5′-catcaatgacgccaaggatgtg-3′ and reverse 5′-cacatccttggcgtcattgatg-3′; primer set for Mut4, forward 5′-gacttcaaggctgtgacgcac-3′ and reverse 5′-gtgcgtcacagccttgaagtc-3′; primer set for Mut5, forward 5′-catcaatgacgccaaggctgtgacgcac-3′ and reverse 5′-gtgcgtcacagccttggcgtcattgat-3′. pCDNA3/ALG-2^WT^ RNAiR [73], pCAG-Myc/TSG101, pCAG-Myc/VPS28, pCAG-Myc/MVB12A, pCAG-Myc/MVB12B, pCAG-Myc/VPS37A, pCAG-Myc/VPS37B, pCAG-Myc/VPS37C, pCAG-Myc/VPS37D [12,43] were constructed as described previously.

### 4.3. Cell Culture and DNA Transfection

HEK293 cells, ALG-2 knockout (KO) HEK293 cells [38], HEK293T cells, MCF-7 cells were cultured in DMEM (Nissui, Tokyo, Japan) supplemented with 5% (for HEK293 cells and HEK293T cells) or 10% (for MCF-7 cells) fetal bovine serum (FBS), 4 mM _L_-glutamine, 100 U/mL penicillin and 100 μg/mL streptomycin at 37 °C under humidified air containing 5% CO_2_. One day after cells were seeded and cultured, cells were then transfected with the expression plasmid DNAs by using the conventional PEI-MAX method (24764, Polysciences Inc., Warrington, PA, USA).

### 4.4. Establishment of MCF-7 Cells with Inducible CDIP1 Expression

Tet-On system-induced MCF-7 cells were generated by infection of lentivirus that is responsible for transcriptional activation of target gene controlled by tetracycline (TET) or doxycycline (DOX). To construct a plasmid DNA that possesses the TET response elements for CDIP1, the DNA fragment encoding CDIP1 was amplified from pcDNA3/CDIP1 with a pair of primer: forward 5′-gtaccgcgggcccgggatccgcgaagatgtccagcgagcctcc-3′ and reverse 5′-acaagaaagctgggtctagattagcacaggcgcttgtacg-3′. The product was inserted between the BamHI and XbaI sites of pLenti CMV TRE3G eGFP Puro using a SLiCE. pLenti CMV TRE3G eGFP Puro (Addgene plasmid no. w819-1) was a gift from Dr. Eric Campeau. The resultant plasmid was designated pLenti CMV TRE3G CDIP1 Puro. For virus packaging, HEK293T cells were seeded and cultured one day and then co-transfected with respective lentivectors pLenti CMV rtTA3 Blast (Addgene plasmid no. w756-1) or pLenti CMV TRE3G CDIP1 Puro in combination with packaging plasmid psPAX2and envelope plasmid pMD2.G by using FuGENE 6 (Promega, Madison, WI, USA). pMD2.G (Addgene plasmid no. 12259) and psPAX2 (Addgene plasmid no. 12260) were gifts from Dr. Didier Trono. Two day after transfection, virus-containing media was collected, centrifuged to remove the cell debris, and then filtered through a 0.45 μm filter (Advantech, Tokyo, Japan), aliquoted into 1.5 mL-tubes and stored in −80 °C before titration or infection. For titrating, MCF-7 cells were seeded and cultured one day. Viruses were serially diluted and then used to infect the cells with 10 μg/mL polybrene (Nacalai Tesque, Kyoto, Japan). The next day, medium was changed to fresh medium containing antibiotics (10 μg/mL blasticidin or 1 μg/mL puromycin). After cultured for appropriate selection duration, cells were trypsinized, harvested and counted. Then MCF-7 cell line including DOX-inducible expression system of CDIP1 was established in two steps: first, parental MCF-7 cells were infected with the lentivirus particles derived from pLenti CMV rtTA3 Blast and selected with blasticidin; next, the cells were infected with the viruses from pLenti CMV TRE3G CDIP1 Puro and selected with puromycin and blasticidin. Each infection was conducted at multiplicity of infection (MOI) of 1. The established cell line was designated MCF-7 Tet-On CDIP1 and cultured in DMEM supplemented with 10% FBS, 4 mM _L_-glutamine, 100 U/mL penicillin, 100 μg/mL streptomycin, 1 μg/mL puromycin and 10 μg/mL blasticidin at 37 °C under humidified air containing 5% CO_2_. DOX (final concentration 100 ng/mL) were added in the culture medium to induce CDIP1 expression.

### 4.5. Immunoprecipitaion Assays

HEK293 cells transiently transfected with expression plasmids for SGFP2, SGFP2-fused proteins and Myc-tagged protein were harvested with phosphate-buffered saline (PBS) (137 mM NaCl, 2.7 mM KCl, 8 mM Na_2_HPO_4_ and 1.5 mM KH_2_PO_4_, pH 7.4), lysed with lysis buffer HKM (20 mM HEPES-NaOH, pH 7.4, 142.5 mM KCl, 1.5 mM MgCl_2_) containing 0.2% Triton X-100, protease inhibitors (1 μM E-64, 3 μg/mL leupeptin, 0.1 mM Pefabloc, 1 μM pepstatin A, 0.2 mM phenylmethanesulfonyl fluoride) and phosphatase inhibitors (10 mM β-glycerophosphate, 50 mM NaF, 1 mM Na_3_VO_4_) and incubated on ice for 30 min. After centrifugation at 18,000× *g* for 10 min at 4 °C, the obtained cell lysates were incubated with glutathione-S-transferase (GST)-fused GFP nanobody prebound to glutathione Sepharose 4B beads (GE Healthcare, Uppsala, Sweden) in the presence of 5 mM EGTA or 100 μM (Figure 1B,D and Figure 4C), 10 µM (Figure 4A,B and Figure 8) or 1 µM (Figure 5) CaCl_2_ overnight at 4 °C. The expression plasmid for GST-GFP nanobody was kindly provided by Drs. Nakayama and Katoh, Kyoto University [74]. The immunocomplexes were washed three times with buffer HKM containing 0.1% Triton X-100 in combination with EGTA or CaCl_2_. The IP products were prepared for SDS-PAGE by boiling at 95 °C in SDS sample buffer (62.5 mM Tris-HCl pH 6.8, 10% glycerol, 2% SDS) supplemented with 0.002% bromophenol blue (BPB) and 5% 2-mercaptoethanol (2-ME). Proteins in the cleared cell lysate (Input) and the IP product were analyzed by WB with the indicated antibodies.

HEK293 cells were treated with BFA (0.2 µg/mL) for 9 h and they were harvested with PBS, lysed with the buffer HKM containing 1% CHAPS, protease inhibitors and phosphatase inhibitors as above and incubated on ice for 30 min. After sonication followed by centrifugation at 18,000× *g* for 10 min, the obtained cleared cell lysates were incubated with rabbit polyclonal antibody against ALG-2 or control rabbit IgG on ice for 1 h followed by incubation in the presence of 5 mM EGTA or 10 μM CaCl_2_ overnight at 4 °C. For IP with rabbit polyclonal antibody against VAPB, the cleared cell lysates were incubated in the presence of 5 mM EGTA or 10 µM CaCl_2_ for 30 min prior to addition of the antibody. The immunocomplexes were collected by incubation with Protein G magnetic beads (Dynabeads^®^ protein G, Invitrogen, Carlsbad, CA, USA) for 1 h at 4 °C. The immunocomplexes were washed three times with buffer HKM containing 0.3% CHAPS and 5 mM EGTA or 10 μM CaCl_2_. Proteins in the immunoprecipitates were eluted with elution buffer (50 mM Glycine-HCl, pH 3) containing 0.3% CHAPS at room temperature for 3 min. After neutralization by adding 1 M Tris-HCl, pH 8.5, the immunoprecipitated proteins were prepared for SDS-PAGE by boiling in SDS sample buffer supplemented with BPB and 2-ME. For WB with antibody against CDIP1, the protein samples were incubated at 37 °C for 1 h in SDS sample buffer without 2-ME.

MCF-7 Tet-On CDIP1 cells were treated with or without DOX (100 ng/mL) for 24 h and they were harvested with PBS, lysed with buffer HKM containing 1% CHAPS, protease inhibitors and phosphatase inhibitors as above and incubated on ice for 30 min. After sonication followed by centrifugation at 18,000× *g* for 10 min, the obtained cleared cell lysates were incubated with rabbit polyclonal antibody against CDIP1 or control rabbit IgG in the presence of 5 mM EGTA or 10 µM CaCl_2_ overnight at 4 °C followed by incubation with Protein G magnetic beads for 1 h at 4 °C. The beads were washed three times with the buffer HKM containing 0.3% CHAPS and 5 mM EGTA or 10 µM CaCl_2_. Proteins in the immunocomplexes were eluted with elution buffer for 3 min at room temperature and processed for sample preparation as above.

### 4.6. Cell Death Assays

One day after HEK293 cells or ALG-2 KO HEK293 cells had been seeded in 3 cm dishes or 6-well plates, they were transfected with the expression plasmids by using PEI-MAX. After 40 h (Figure 3) or 24 h (Figure 6 and Figure 9), cell mortality was measured by quantifying the amount of lactate dehydrogenase (LDH) released from dead cells using the CytoTox96^®^ Non-Radioactive Cytotoxicity Assay (Promega, Madison, WI, USA) according to the manufacturer’s instruction. Briefly, aliquots of medium (each 30 μL) were incubated with equal volume of CytoTox96^®^ reagent at room temperature for 30 min. Then, 30 μL of the stop solution was added to the mixture and absorbance at 490 nm was measured. The total amount of LDH in the medium was normalized by dividing by whole protein amount of the cell lysate in each dish or well that had been measured by using the Pierce™ BCA Protein Assay Kit (Thermo Fisher Scientific, Waltham, MA, USA). Cytotoxicity was expressed as fold LDH release compared with mock-transfected cells. To verify the amount of GFP-fused and Myc-tagged proteins, cells were lysed in SDS sample buffer containing protease inhibitors as above. The cleared cell lysates were obtained by sonication and centrifugation, and then processed for SDS-PADE sample preparation. The activity of caspase-3/7 was measured using the Caspase-Glo^®^ 3/7 Assay System (Promega, Madison, WI, USA) according to the manufacturer’s instructions. Cells were harvested with PBS, lysed with lysis buffer TNE (20 mM Tris-HCl, pH 7.5, 120 mM NaCl, 1 mM EDTA) containing 1% Triton X-100, 3 μg/mL leupeptin, 1 μM pepstatin A and 0.2 mM phenylmethanesulfonyl fluoride, and incubated on ice for 30 min. After centrifugation at 14,000× *g* for 10 min, the protein amount of the obtained cleared cell lysates was measured by using the BCA Protein Assay Kit. Then, equal amounts of total cellular protein lysates were subjected to the caspase-3/7 assay.

### 4.7. Exploration of CDIP1-Interacting Proteins

Two days after MCF-7 Tet-On CDIP1 cells had been seeded in 100 mm dishes, they were treated with or without DOX (100 ng/mL) for 24 h, harvested, and lysed with the buffer HKM containing 1% CHAPS, protease inhibitors and phosphatase inhibitors as above. The cleared cell lysates were obtained by sonication and centrifugation at 18,000× *g* for 10 min and incubated with rabbit polyclonal antibody against CDIP1 or control rabbit IgG at 4 °C overnight, followed by incubation with Protein G magnetic beads for 2 h. The beads were washed three times with the buffer HKM containing 0.3% CHAPS and then incubated in elution buffer at room temperature for 3 min to elute proteins in the immunocomplexes. After adding 1 M Tris-HCl, pH 8.5, for neutralization, the bound proteins were subjected to SDS-PAGE, followed by silver staining. For mass spectrometry, the samples were loaded to SDS-PAGE gel, resolved about 2.0 cm by the running gel and manually cut into one slice. The gels were fixed with 50% methanol/water containing 5% acetic acid, and the fixed gels were washed three times with pure water and subjected to in-gel digestion using Trypsin/Lys-C Mix (Promega, Madison, WI, USA). The resultant peptides were analyzed by nano-flow reverse phase liquid chromatography followed by tandem MS, using a Q Exactive Hybrid Quadrupole-Orbitrap Mass Spectrometer (Thermo Fisher Scientific, Waltham, MA, USA).

### 4.8. Chromatographic and Mass Spectroscopic and Methods, Instrumentations and Database Searches

A capillary reverse phase HPLC-MS/MS system composed of a Dionex U3000 gradient pump equipped with VICI CHEMINERT valve, and Q Exactive equipped with a nano-electrospray ionization (NSI) source (AMR, Tokyo, Japan). The desalted peptides were loaded into a separation capillary C18 reverse phase column (NTCC-360/100-3-125, 125 × 0.1 mm, Nikkyo Technos, Tokyo, Japan). Xcalibur system software vervion 3.0.63 (Thermo Fisher Scientific, Waltham, MA, USA) was used to record peptide spectra over the mass range of *m*/*z* 350–1800. Repeatedly, MS spectra were recorded followed by 10 data-dependent high energy collisional dissociation (HCD) MS/MS spectra generated from 10 highest intensity precursor ions. Multiple charged peptides were chosen for MS/MS experiments due to their good fragmentation characteristics. MS/MS spectra were interpreted and peak lists were generated by Proteome Discoverer software version 2.2.0.388 (Thermo Fisher Scientific, Waltham, MA, USA). Searches were performed by using the SEQUEST (Thermo) against *Homo sapiens* (SwissProt TaxID = 9606) peptide sequence. Peptide identifications were based on significant Xcorr (high confidence filter). Peptide identification and modification information returned from SEQUEST were manually inspected and filtered to obtain confirmed peptide identification and modification lists of HCD MS/MS.

### 4.9. Statistical Analysis

Statistical analysis was performed by one-way analysis of variance (ANOVA) followed by Tukey’s test using Origin 9.1 (Micro Software, Northampton, MA, USA). *p*-values less than 0.05 are considered statistically significant.

### 4.10. Research Ethics

We followed biosafety guidelines for recombinant DNA research at Nagoya University. Experimental proposals were approved by the Recombinant DNA Biosafety Committee of the Graduate School of Bioagricultural Sciences, Nagoya University: Nou15-067 (approved on 24 March 2006) and Nou19-002 (approved on 12 April 2019).

## Figures and Tables

**Figure 1 ijms-22-01175-f001:**
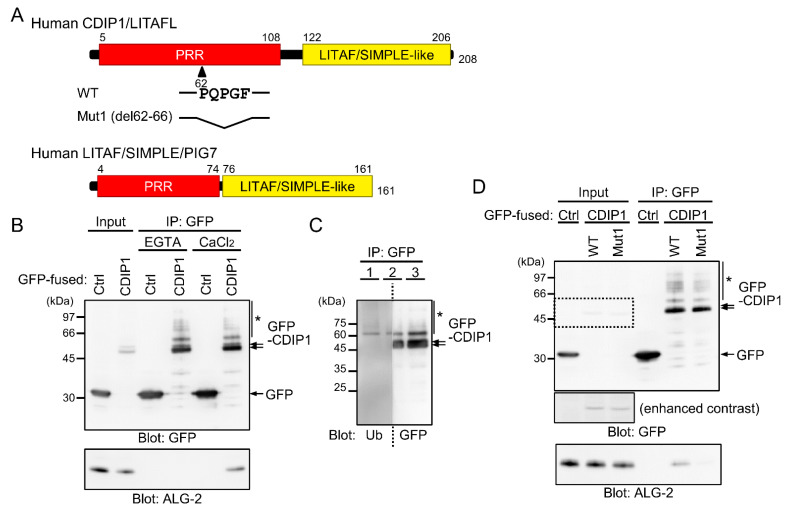
Ca^2+^-dependent interaction between apoptosis-linked gene 2 (ALG-2) and cell death-inducing p53 target protein 1 (CDIP1). (**A**) Schematic diagrams of human CDIP1/Lipopolysaccharide-induced tumor necrosis factor-α factor-like (LITAFL) and human Lipopolysaccharide-induced tumor necrosis factor-α factor (LITAF)/Small integral membrane protein of lysosome/late endosome (SIMPLE)/p53-induced gene 7 protein (PIG7) showing the N-terminal Pro-rich region (PRR) and the C-terminal LITAF/SIMPLE-like domain. CDIP1 contains an ALG-2 binding motif (ABM)-2-like motif (^62^-PQPGF) in the PRR. (**B**) Co-immunoprecipitation (Co-IP) assay using HEK293 cells transiently expressing green fluorescent protein (GFP) (Ctrl) or GFP-fused CDIP1 (CDIP1) was performed as described in Materials and Methods. GST-fused GFP nanobody (GST-GFP nanobody) pre-bound to glutathione beads was added to the cleared cell lysate (Input) in the presence of 5 mM EGTA (EGTA) or 100 μM CaCl_2_ (CaCl_2_), and cleared cell lysate proteins (Input) and immunoprecipitated proteins (IP) were analyzed by SDS-PAGE followed by Western blot (WB) with antibodies against GFP (upper panel) and ALG-2 (lower panel). The relative amounts of cleared cell lysate proteins (Input) used for analysis of IP products (IP) were 20% (upper panel) and 2.5% (lower panel). (**C**) Ubiquitination of GFP-CDIP1. IP products obtained with the GFP nanobody from HEK293 cells transiently expressing GFP-CDIP1 were resolved in three lanes (1 to 3) of an SDS-PAGE gel and the blotted membrane was cut into halves in the middle of the second lane. The left and right membranes were probed with monoclonal antibodies against ubiquitin (Ub) and GFP (GFP), respectively. Ubiquitinated GFP-CDIP1 bands are marked with asterisks. (**D**) HEK293 cells transiently expressing GFP (Ctrl), GFP-CDIP1 (WT) or GFP-CDIP1 Mut1 (Mut1) and IP products (IP) obtained with the GFP nanobody in the presence of 100 μM CaCl_2_ were subjected to SDS-PAGE followed by immunoblotting with antibodies against GFP (upper panel) and ALG-2 (lower panel). Contrast of the original image was decreased (−40%) to clearly show the bands for GFP-CDIP1 in the input lanes (enhanced contrast). The relative amounts of cleared cell lysate proteins (Input) used for analysis of IP products (IP) were 20% (upper panel) and 2.5% (lower panel). Arrows, non-ubiquitinated proteins; asterisks, slower-migrating ubiquitinated proteins.

**Figure 2 ijms-22-01175-f002:**
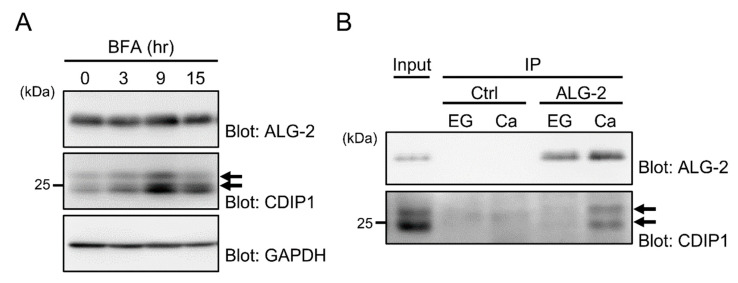
Interaction between ALG-2 and CDIP1 in cells treated with BFA. (**A**) BFA-induced expression of CDIP1. HEK293 cells were treated with 0.2 μg/mL BFA for the indicated time, and the cell lysates were subjected to SDS-PAGE followed by immunoblotting with antibodies against ALG-2, CDIP1 and GAPDH. Doublet bands were detected by the antibody against CDIP1. (**B**) Co-IP assay of BFA-treated HEK293 cells. Cells were treated with BFA for 9 h, and the cleared cell lysate was immunoprecipitated with a polyclonal antibody against ALG-2 (ALG-2) or control IgG (Ctrl) in the presence of 5 mM EGTA (EG) or 10 μM CaCl_2_ (Ca). Cleared cell lysate proteins (Input) and immunoprecipitated proteins (IP) were subjected to SDS-PAGE followed by immunoblotting with antibodies against ALG-2 and CDIP1. The relative amounts of cleared cell lysate proteins (Input) used for analysis of IP products (IP) were 10% (upper panel) and 0.5% (lower panel). Arrows, non-ubiquitinated CDIP1 proteins.

**Figure 3 ijms-22-01175-f003:**
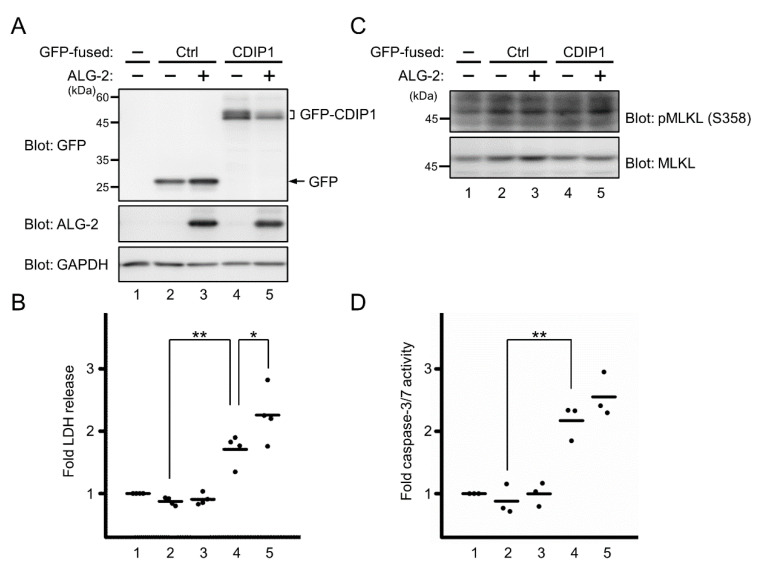
Promotion of CDIP1-induced cell death by ALG-2 expression. ALG-2 KO HEK293 cells were transfected with expression plasmids for GFP (Ctrl) or GFP-CDIP1 (CDIP1) in combination with or without expression plasmids for ALG-2. Mock-transfected cells were used as a reference to calculate the relative release of lactate dehydrogenase (LDH) and the intracellular caspase-3/7 activity. (**A**) Representative results of WB analysis of protein levels of GFP-CDIP1 (upper panel) and ALG-2 (middle panel) in cells at 40 h after transfection. GAPDH was detected as a loading control (lower panel). (**B**) Cell death was monitored by measuring the amounts of LDH released into the medium. Values are expressed as fold LDH release into the medium compared with that of mock-transfected cells. Four independent repetitive experiments were performed and data are expressed as dot plots with means (bar). Statistical significance by one-way ANOVA followed by Tukey’s test is indicated by asterisks (** *p* < 0.01 and * *p* < 0.05). (**C**) Representative results of WB analysis of protein levels of phosphorylated MLKL (upper panel) and total MLKL (lower panel) in cells at 40 h after transfection. (**D**) Caspase-3/7 activity was measured using the Caspase-3/7-Glo assay system at 40 h after transfection. Values are expressed relative to mock-transfected cells. Three independent repetitive experiments were performed and data are expressed as dot plots with means (bar). Statistical significance by one-way ANOVA followed by Tukey’s test is indicated by asterisks (** *p* < 0.01).

**Figure 4 ijms-22-01175-f004:**
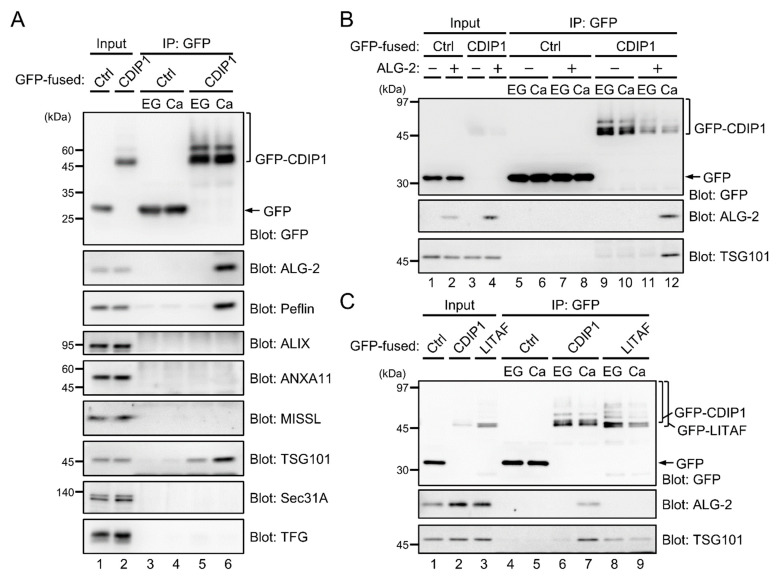
Ca^2+^-dependent adaptor function of ALG-2 bridging between CDIP1 and TSG101. (**A**) Screening of proteins interacting with CDIP1 in a Ca^2+^-dependent manner. Cleared cell lysates of HEK293 cells transiently expressing GFP (Ctrl) or GFP-CDIP1 (CDIP1) and IP products (IP) obtained with the GFP nanobody in the presence of 5 mM EGTA (EG) or 10 μM CaCl_2_ (Ca) were subjected to SDS-PAGE followed by immunoblotting with antibodies against the indicated proteins. The relative amounts of cleared cell lysate proteins (Input) used for analysis of IP products (IP) were 20% (upper panel, blot for GFP) and 0.5% (other panels). (**B**) Enhancement of the interaction between CDIP1 and TSG101 by ALG-2 and Ca^2+^. ALG-2 KO HEK293 cells were transiently transfected with expression plasmids for GFP (Ctrl) or GFP-CDIP1 (CDIP1) in combination with or without expression plasmids for ALG-2. Cleared cell lysates (Input) and immunoprecipitation products (IP) obtained with the GFP nanobody in the presence of 5 mM EGTA (EG) or 10 μM CaCl_2_ (Ca) were subjected to SDS-PAGE followed by immunoblotting with antibodies against GFP, ALG-2 and TSG101. The relative amounts of cleared cell lysate proteins (Input) used for analysis of IP products (IP) were 10% (upper panel) and 2% (middle and lower panels). (**C**) Lack of interaction between LITAF/SIMPLE and ALG-2. Cleared cell lysates of HEK293 cells transiently expressing GFP (Ctrl), GFP-CDIP1 (CDIP1) or GFP-LITAF/SIMPLE (LITAF) and IP products (IP) obtained with the GFP nanobody in the presence of 5 mM EGTA (EG) or 100 μM CaCl_2_ (Ca) were subjected to SDS-PAGE followed by immunoblotting with antibodies against GFP, ALG-2 and TSG101. The relative amounts of cleared cell lysate proteins (Input) used for analysis of IP products (IP) were 10% (upper panel) and 2% (middle and lower panels).

**Figure 5 ijms-22-01175-f005:**
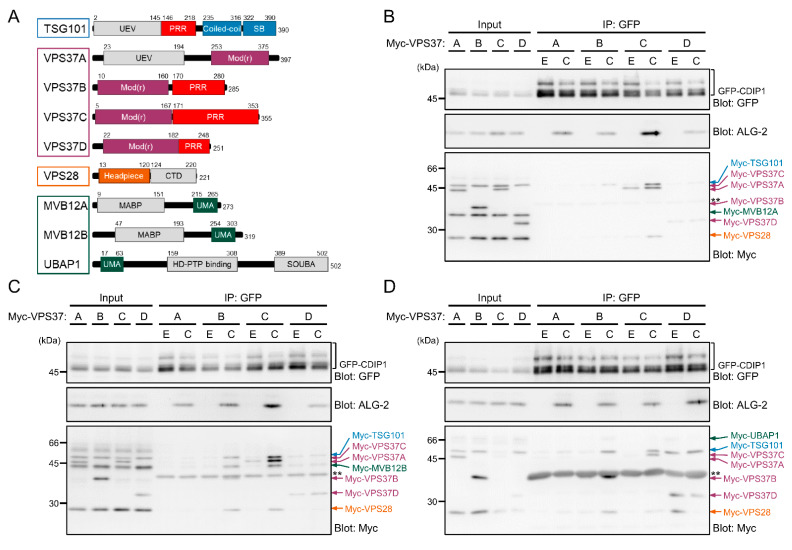
Different Ca^2+^ dependence on interaction of CDIP1 with ESCRT-I isocomplexes. (**A**) Schematic diagrams of ESCRT-I subunits. UEV, ubiquitin E2 variant domain; PRR, Pro-rich region; SB, steadiness box; Mod(r), modifier of rudimentary domain; CTD, C-terminal domain; MABP, MVB12-associated β-prism domain; UMA, UBAP1-MVB12-associated domain; SOUBA, solenoid of overlapping UBAs domain. ALG-2 directly binds to PRRs of TSG101, VPS37B and VPC37C in the presence of Ca^2+^ [12,45]. (**B**–**D**) ALG-2 KO HEK293 cells were transiently transfected with expression plasmids for GFP-CDIP1, ALG-2, Myc-TSG101, Myc-VPS28 and each isoform of VPS37A-D in combination with an expression plasmid for Myc-MVB12A (**B**), Myc-MVB12B (**C**) or Myc-UBAP1 (**D**). Cleared cell lysates (Input) and IP products (IP) obtained with the GFP nanobody in the presence of 5 mM EGTA (E) or 1 μM CaCl_2_ (**C**) were subjected to SDS-PAGE followed by immunoblotting with antibodies against GFP, ALG-2 and Myc tag. The relative amounts of cleared cell lysate proteins (Input) used for analysis of IP products (IP) were 10% (upper panel), 1.3% (middle panel) and 2% (lower panel). A double asterisk (**) indicates the migration position of GST-fused GFP nanobody in the IP products.

**Figure 6 ijms-22-01175-f006:**
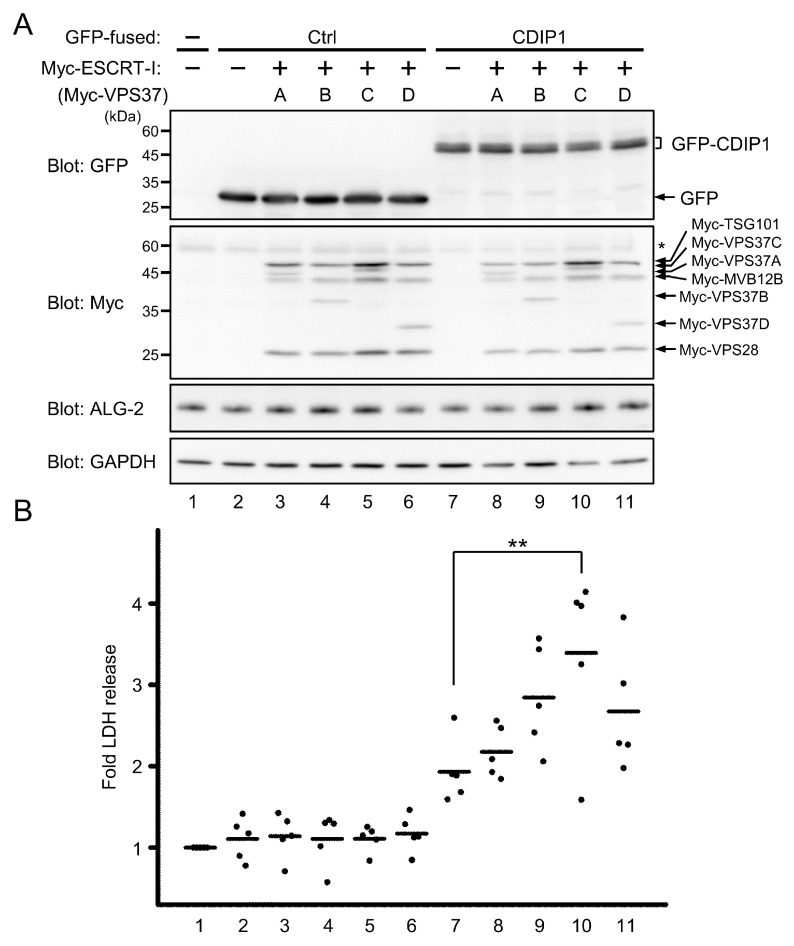
Promotion of CDIP1-induced cell death by ESCRT-I isocomplexes. HEK293 cells were transfected with expression plasmids for GFP (Ctrl) or GFP-CDIP1 (CDIP1) in combination with or without expression plasmids for Myc-tagged ESCRT-I subunits (Myc-TSG101, Myc-VPS28, Myc-MVB12B and each isoform of VPS37A-D). Mock-transfected cells were used as a reference to calculate the relative release of lactate dehydrogenase (LDH). (**A**) Representative results of WB analysis of protein levels of GFP-CDIP1, Myc-tagged proteins and ALG-2 in cells at 24 h after transfection. GAPDH was detected as a loading control. A single asterisk (*) indicates the migration position of the endogenous Myc protein. (**B**) Cell death was monitored by measuring the amounts of LDH released into the medium. Values are expressed as fold LDH release into the medium compared with that of mock-transfected cells. Five independent repetitive experiments were performed and data are expressed as dot plots with means (bar). Statistical significance by one-way ANOVA followed by Tukey’s test is indicated by asterisks (** *p* < 0.01).

**Figure 7 ijms-22-01175-f007:**
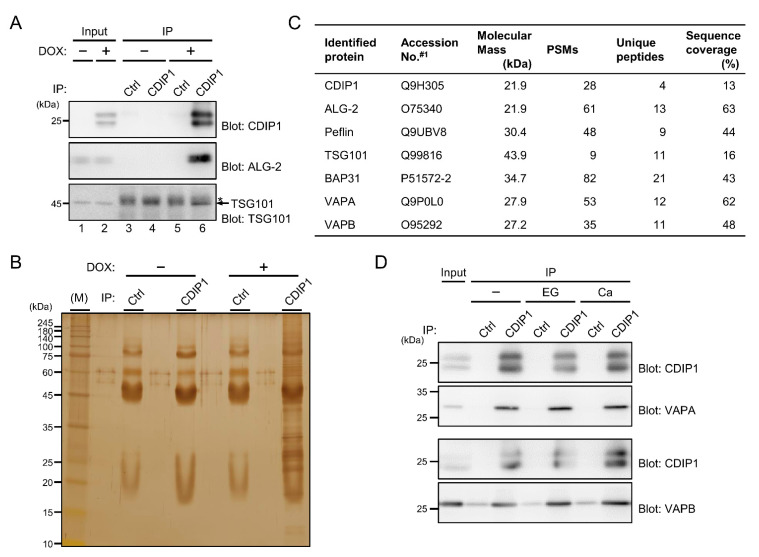
Interaction of CDIP1 with VAPA and VAPB. (**A**,**B**) Co-IP analysis of MCF-7 Tet-On CDIP1 cells cultured in the absence (−) or presence (+) of the tetracycline analogue doxycycline (DOX). Cleared cell lysates (Input) and IP products (IP) obtained with an antibody against CDIP1 (CDIP1) or control IgG (Ctrl) were subjected to SDS-PAGE followed by immunoblotting with antibodies against CDIP1, ALG-2 and TSG101 (**A**). The relative amounts of cleared cell lysate proteins (Input) used for analysis of IP products (IP) were 20% (upper panel) and 1% (middle and lower panels). An asterisk indicates the migration position of IgG heavy chain proteins. IP products obtained with the antibody against CDIP1 (CDIP1) or control IgG (Ctrl) were subjected to SDS-PAGE followed by silver staining (**B**). (**C**) List of identified proteins in the IP product of the antibody against CDIP1 from cells cultured in the presence of DOX. #1, accession numbers in the UniProt database. (**D**) MCF-7 Tet-On CDIP1 cells were cultured in the presence of DOX. Cleared cell lysates (Input) and IP products (IP) obtained with the antibody against CDIP1 (CDIP1) or control IgG (Ctrl) in the absence (−) or presence of 5 mM EGTA (EG) or 10 μM CaCl_2_ (Ca) were subjected to SDS-PAGE followed by immunoblotting with antibodies against CDIP1, VAPA and VAPB. The relative amounts of cleared cell lysate proteins (Input) used for analysis of IP products (IP) were 20% (blot for CDIP1) and 1% (blot for VAPA and VAPB).

**Figure 8 ijms-22-01175-f008:**
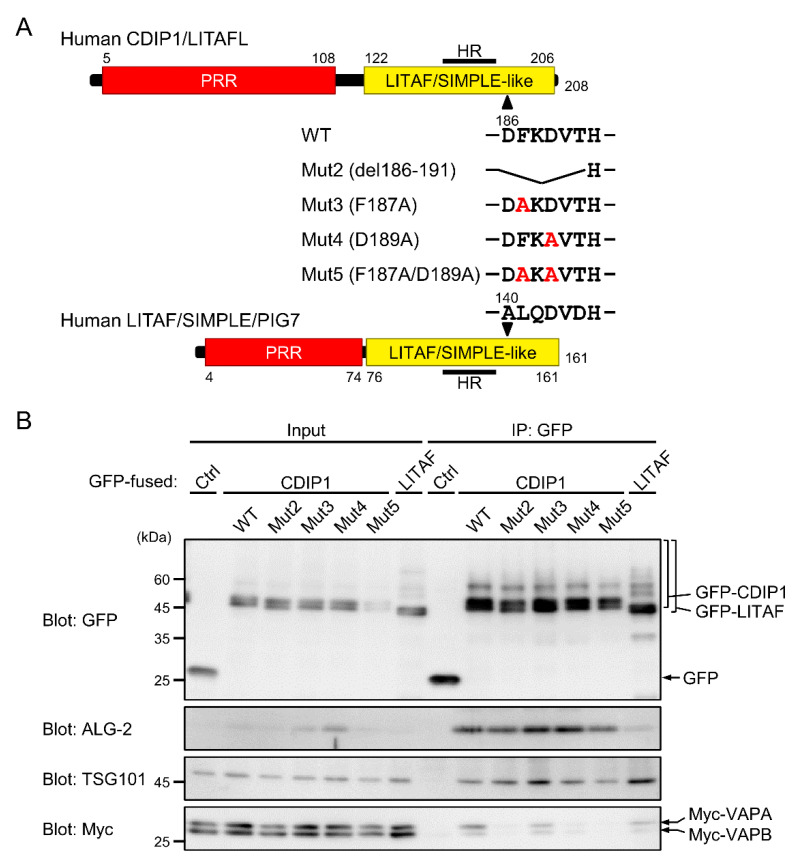
Effects of mutations in the FFAT-like motif on the interaction of CDIP1 with VAPA, VAPB, ALG-2 and TSG101. (**A**) Schematic diagram of mutations at the FFAT-like motif in the C-terminal LITAF/SIMPLE-like domain of CDIP1. HR, hydrophobic region. (**B**) HEK293 cells were transiently transfected with expression plasmids for Myc-VAPA and Myc-VAPB in combination with expression plasmids for GFP (Ctrl), GFP-CDIP1 (WT), GFP-CDIP1 mutants (Mut2–5) or GFP-LITAF/SIMPLE (LITAF). Cleared cell lysates (Input) and IP products (IP) obtained with the GFP nanobody in the presence of 10 μM CaCl_2_ were subjected to SDS-PAGE followed by immunoblotting with antibodies against GFP, ALG-2, TSG101 and Myc tag. The relative amounts of cleared cell lysate proteins (Input) used for analysis of IP products (IP) were 10% (upper panel) and 1.3% (other panels).

**Figure 9 ijms-22-01175-f009:**
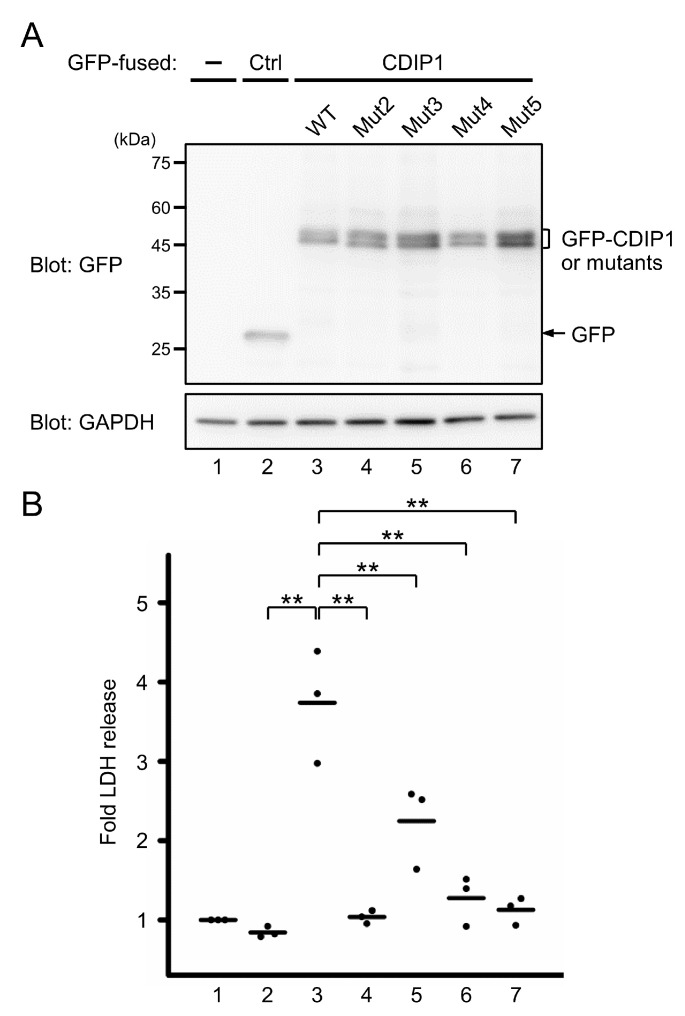
Effects of mutations in the FFAT-like motif on CDIP1-induced cell death. HEK293 cells were transfected with expression plasmids for GFP (Ctrl), GFP-CDIP1 (WT) or GFP-CDIP1 mutants (Mut2–5). Mock-transfected cells were used as a reference to calculate the relative release of lactate dehydrogenase (LDH). (**A**) Representative results of WB analysis of levels of GFP and GFP-fused proteins in cells at 24 h after transfection. GAPDH was detected as a loading control. (**B**) Cell death was monitored by measuring the amounts of LDH released into the medium. Values are expressed as fold LDH release into the medium compared with that of mock-transfected cells. Three independent repetitive experiments were performed and data are expressed as dot plots with means (bar). Statistical significance by one-way ANOVA followed by Tukey’s test is indicated by asterisks (** *p* < 0.01).

**Figure 10 ijms-22-01175-f010:**
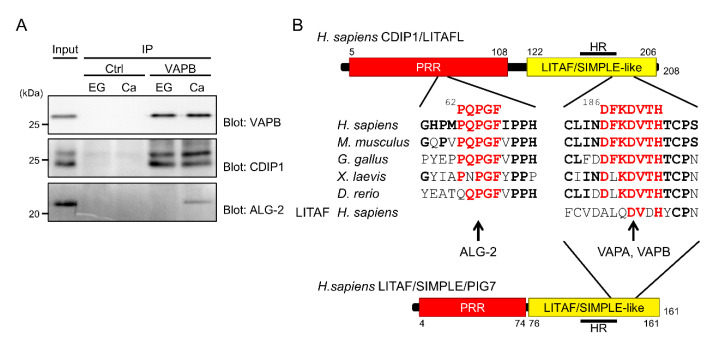
Ca^2+^-dependent ternary complex formation between VAPB, CDIP1 and ALG-2. (**A**) Co-IP assay of BFA-treated HEK293 cells. Cells were treated with BFA for 9 h, and the cleared cell lysate was immunoprecipitated with a polyclonal antibody against VAPB (VAPB) or control IgG (Ctrl) in the presence of 5 mM EGTA (EG) or 10 μM CaCl_2_ (Ca). Cleared cell lysates (Input) and IP products (IP) were subjected to SDS-PAGE followed by immunoblotting with antibodies against VAPB, CDIP1 and ALG-2. The relative amounts of cleared cell lysate proteins (Input) used for analysis of IP products (IP) were 40% (upper panel) and 0.5% (middle and lower panels). (**B**) Alignment of the ABM-2-like motif and the FFAT-like motif of CDIP1 homologs from humans (Q9H305), mice (Q9DB75), chickens (A0A1D5PGK0), flogs (Q8AVW3) and zebrafish (A6H8U1) and the FFAT-like motif of human LITAF (Q99732). Residues conserved among CDIP1 homologs and LITAF are indicated in red. HR, hydrophobic region.

## Data Availability

Not applicable.

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
