# Peer review of "The Novel ALG-2 Target Protein CDIP1 Promotes Cell Death by Interacting with ESCRT-I and VAPA/B"

_ijms, 2021, doi:10.3390/ijms22031175_

Round 1

Reviewer 1 Report

The paper under review is entitled “The Novel ALG-2 Target Protein CDIP1 Promotes Cell Death 2 by Interacting with ESCRT-I and VAPA/B”. Using mainly Co-IP authors demonstrate that ALG-2 (Apoptosis-linked gene 2) protein interacts with CDIP1 protein (cell death-inducing p53 target protein 1) in calcium-dependent manner and thus enables interaction of CDIP1 with ESCRT-I complex (endosomal sorting complex required for transport-I) and VAPA/B protein (vesicle-associated membrane protein A and B). Further authors demonstrate that overexpression of CDIP1 protein in ALG2-KO HEK293T cells results in cell death measured by LDH activity (lactate dehydrogenase) in extracellular medium. This effect was enhanced when ALG-2 was also expressed. Co-expression of CDIP1 protein and ESCRT-I subunits in WT HEK293T cells further exacerbated toxic effect of CDIP1, whereas disruption of interaction between CPIP1 and VAPA/B by mutating FFAT-motif of CDIP1 reduced CDIP1 toxicity in HEK293T cells. Overall the study is of interest and present how ALG-2 protein promotes CDIP1-induced cell death by promoting its interaction with ESCRT I complex and VAPA/B protein; however, clarification of few points could make it better: 1) In Figure 7A it is not clear that TSG101 precipitates only with CDIP1 and DOX + conditions. 2) Authors paid much attention to calcium dependence of interaction between ALG-2, CDIP-1, VAPA/B and ESCRT-I complex. It might be beneficial for the manuscript to clarify why calcium conditions for Co-IP varied and what is functional significance for calcium regulation of interaction between ALG-2 and CDIP-1 protein. I hope the authors find this comments useful and it will help to improve the manuscript.

Author Response

Response to Reviewer 1 Comments

The paper under review is entitled “The Novel ALG-2 Target Protein CDIP1 Promotes Cell Death 2 by Interacting with ESCRT I and VAPA/B”. Using mainly Co IP authors demonstrate that ALG-2 (Apoptosis linked gene 2) protein interacts with CDIP1 protein (cell d eath inducing p53 target protein 1) in calcium dependent manner and thus enables interaction of CDIP1 with ESCRT I complex (endosomal sorting complex required for transport I) and VAPA/B protein (vesicle associated membrane protein A and B). Further author s demonstrate that overexpression of CDIP1 protein in ALG2 KO HEK293T cells results in cell death measured by LDH activity (lactate dehydrogenase) in extracellular medium. This effect was enhanced when ALG-2 was also expressed. Co expression of CDIP1 prote in and ESCRT I subunits in WT HEK293T cells further exacerbated toxic effect of CDIP1, whereas disruption of interaction between CPIP1 and VAPA/B by mutating FFAT motif of CDIP1 reduced CDIP1 toxicity in HEK293T cells. Overall the study is of interest and present how ALG 2 protein promotes CDIP1 induced cell death by promoting its interaction with ESCRT I complex and VAPA/B protein; however, clarification of few points could make it better:

Response: Thank you very much for positive comments and kind advi ce to improve the manuscript. We revised the manuscript according to suggestions given by you and by Reviewer 2.

1) In Figure 7A it is not clear that TSG101 precipitates only with CDIP1 and DOX +
conditions.

Response: As yo u pointed out, we agree that th e band for TSG101 in the
immunoprecipitate of antibodies against CDIP1 from cells treated with DOX was not clear, since the bands corresponding to the heavy chain s of immunoglobulin G (IgG-H)were detected with higher intensity than the band for TSG101. The band s for IgG-H were also detected in other three IP products with similar intensities The band for TSG101 in the IP product of CDIP1 from cells treated with D OX was detected as slightly faster migrating band compared with the band for IgG-H . Therefor e, we believed that TSG101 was only co immunoprecipitated with CDIP1 from the lysate of D OX treated cells. We commented this in the revised manuscript (Page 11, lanes 368 372) and marked the migrating positions o f TSG101 and IgG in Figure 7A

2) Authors paid much attention to calcium dependence of interaction between ALG-2, CDIP-1, VAPA/B and ESCRT-I complex. It might be beneficial for the manuscript to clarify why calcium conditions for Co IP varied and what is functional significance for calcium regulat ion of interaction be tween ALG-2 and CDIP-1 protein.

Response: Thank you f or important suggestion s . The sentences Page 3, lanes 114 115 ; Page 4, lanes 168 169; Page 8, lanes 292 294) were added to clarify why different calcium conditions for Co IP were conducted. Several studies demonstrated that intracellular calcium mobilization are important in apoptosis of cells induced by anti cancer drugs.
Therefore, there is a possibility that ALG-2 interacts with CDIP1 in response to
intracellular calcium mobilization in an ti cancer drug treated cancer cells to promote apoptosis. We described this fact and possibility in the revised manuscript (Page 15, lanes 510 514)

I hope the authors find this comments useful and it will help to improve the manuscript.
Response: We thank the reviewer for her/his comments and suggestions, which are helpful to improve our manuscript.

Reviewer 2 Report

This manuscript reports the role of cell death-inducing p53 target protein 1 (CDIP1), known as a pro-apoptotic protein, to promote cell death by interacting with ESCRT-I and VAPA/B. The authors show that CDIP1 interacts with ALG-2 in a Ca2+-dependent manner and that GFP-CDIP1 associates with tumor susceptibility gene 101 (TSG101), a known target of ALG-2 and a subunit of endosomal sorting complex required for transport-I (ESCRT-I). The authors also identify interaction between vesicle-associated membrane protein (VAP) and CDIP1. Although the paper features some interesting aspects, the cell death mechanism is left to assumption due to the lack of characterization.

  • The author should better characterize cell death in figure 3. LDH is a marker of necrosis rather than apoptosis. WB analysis of apoptosis markers and necroptosis markers, including RIP kinases and MLKL, need to be performed. Additionally, FACS analysis can be easily done using AnnexinV/PI staining. These experiments will help the authors to characterize what kind of cell death is driven by the overexpression of CDIP-1 and enhanced by ALG-2.

  • Since ESCRT-I is responsible for recruiting ESCRT-III, which forms the constriction zone just before the cells separate, and has been involved as a protective event preventing MLKL-dependent necroptosis, it would be interesting to connect the ESCRT system with the cell death mechanisms. Therefore, it is important to understand what kind of cell death the authors deal with.

The molecular analysis of the different partners is well done, I have nothing to add concerning this part of the manuscript.

Author Response

Response to Reviewer 2 Comments

This manuscript reports the role of cell death inducing p53 target protein 1 (CDIP1), known as a pro apoptotic protein, to promote cell death by interacting with ESCRT-I and VAPA/B. The authors show that CDIP1 interacts with ALG 2 in a Ca2+ dependent manner and that GFP CDIP1 associates with tumor susceptibility gene 101 (TSG101), a known target of ALG-2 and a subunit of endosomal sorting complex required for transport I (ESCRT-I). The authors also identify interaction between vesicle associated membrane protein (VAP) and CDIP1. Although the paper features some interesting aspects, the cell death mechanism is left to assumption due to the lack of ch aracterization.
Response: Thank you very much for your favorable comments and kind suggestions to improve our manuscript.

The author should better characterize cell death in figure 3. LDH is a marker of
necrosis rather than apoptosis. WB analysis of apoptosis markers and necroptosis markers, including RIP kinases and MLKL, need to be performed. Additionally, FACS analysis can be easily done using AnnexinV/PI staining. These experiments will help the authors to characterize what kind of cell death is d riven by the overexpression of CDIP-1 and enhanced by ALG-2.

Response: Thank you for important suggestions. According to your suggestions, we performed WB analysis of a necroptosis marker, MLKL. As shown in Figure 3C in the revised version , we detected we ak signal for phosphorylated MLKL buried in noise in mock treated cells (lane 1) 1), which was not enhanced by expression of GFP-CDIP1 or co expression of GFP-CDIP1 and ALG-2 . Therefore, GFP-CDIP1 induced cell death does not seem to be necrosis. For us, FACS analysis using Annexin V/PI staining is not so easily conducted, because we expressed GFP or GFP fusion proteins in cells in our experiments and we have no flow cytometry with multiple illumination sources . Instead, we performed luminescent based caspase 3/7 assay by using a Caspase Glo 3/7 assay kit (Promega). As shown in Figure 3D, expression of GFP-CDIP1 resulted in increase of intracellular activity of ca spase 3/7 and this increase appeared to be further enhanced by co-expression with ALG-2. From these results, we concluded that expression of GFP CDIP1 induces apoptosis in HEK293 cells , as shown in other cell lines ref s. 34,36].

Since ESCRT I is responsible for recruiting ESCRT-III, which forms the constriction
zone just before the cells separate, and has been involved as a protective event preventing MLKL dependent necroptosis, it would be interesting to connect the ESCRT sys tem with the cell death mechanisms. Therefore, it is important to understand what kind of cell death the authors deal with.
Response: Thank you for your insightful suggestions. It is unfortunate that we did not find the differences of the level of phospho rylated MLKL between all cells (mock treated cells, cells expressing GFP, GFP-CDIP1, ALG-2 and GFP-CDIP1/ALG-2). Thus, necroptosis may not be induced by expression of GFP-CDIP1. Instead, we found that caspase 3/7 activity was enhanced by expression of GFP-CDIP1, suggesting that GFP-CDIP1 induces apoptosis. We presented the data (Figures 3C and 3D) and described this fact in the revised manuscript (Page 5, lanes 198 211)

The molecular analysis of the different partners is well done, I have nothing to add concerning this part of the manuscript.
Response: We thank the reviewer for her/his valuable comments. According to kind suggestion s of the reviewer, we characterized CDIP1-induced cell death, which is helpful to improve our manuscript .

Round 2

Reviewer 2 Report

the authors have responded accordingly